# Probing the Decision Boundaries of In-context Learning in Large Language Models

**Siyan Zhao, Tung Nguyen, Aditya Grover**
Department of Computer Science
University of California Los Angeles
{siyanz,tungnd,adityag}@cs.ucla.edu

## Abstract

In-context learning is a key paradigm in large language models (LLMs) that enables them to generalize to new tasks and domains by simply prompting these models with a few exemplars without explicit parameter updates. Many attempts have been made to understand in-context learning in LLMs as a function of model scale, pretraining data, and other factors. In this work, we propose a new mechanism to probe and understand in-context learning from the lens of decision boundaries for in-context binary classification. Decision boundaries are straightforward to visualize and provide important information about the qualitative behavior of the inductive biases of standard classifiers. To our surprise, we find that the decision boundaries learned by current LLMs in simple binary classification tasks are often irregular and non-smooth, regardless of linear separability in the underlying task. This paper investigates the factors influencing these decision boundaries and explores methods to enhance their generalizability. We assess various approaches, including training-free and fine-tuning methods for LLMs, the impact of model architecture, and the effectiveness of active prompting techniques for smoothing decision boundaries in a data-efficient manner. Our findings provide a deeper understanding of in-context learning dynamics and offer practical improvements for enhancing robustness and generalizability of in-context learning.[1]

## 1 Introduction

Recent language models, such as GPT-3+ [Brown et al., 2020, Achiam et al., 2023], have demonstrated the ability to scale performance with increased training dataset size and model capacity through the simple objective of next token prediction [Kaplan et al., 2020]. A key emergent behavior of these transformer-based models is in-context learning, which allows the model to learn tasks by conditioning on a sequence of demonstrations without explicit training [Wei et al., 2022]. This unique capability allows LLMs to adapt seamlessly to new tasks, often achieving superior performance in few-shot settings [Brown et al., 2020]. Despite significant successes, the underlying mechanisms of how in-context learning works remain partially understood.

Recent attempts to understand in-context learning have focused on various aspects. From a theoretical standpoint, studies by Von Oswald et al. [2023] and Dai et al. [2023] have linked the mechanisms of in-context learning to gradient descent, suggesting that transformers can emulate optimization processes. On the practical side, research has investigated the impact of different factors on in-context learning. Works by Min et al. [2022b] and Shi et al. [2023] reveal that accurate demonstrations are not essential for effective in-context learning. On the other hand, factors such as the prompt structure and model size [Wei et al., 2023, Webson and Pavlick, 2022], or the order of in-context examples [Chen et al., 2024] greatly affect outcomes. More recently, with the development of LLMs supporting longer

---

[1]Our code is released at `https://github.com/siyan-zhao/ICL_decision_boundary`.

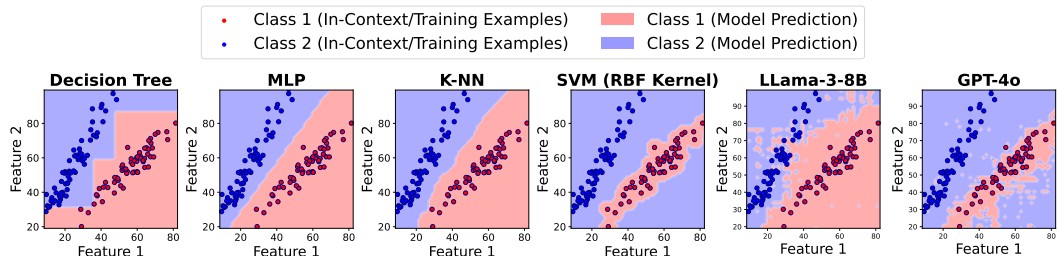

Figure 1: Decision boundaries of LLMs and traditional machine learning models on a linearly separable binary classification task. The background colors represent the model's predictions, while the points represent the in-context or training examples. LLMs exhibit non-smooth decision boundaries compared to the classical models. See Appendix E for model hyperparameters.

context lengths up to 10M [Reid et al., 2024], studies have shown that in-context learning performance improves with significant number of demonstrations [Agarwal et al., 2024, Bertsch et al., 2024], where the performance can be comparable to fine-tuning on the same amount of demonstrations. Additionally, works by Garg et al. [2022], Nguyen and Grover [2022] have demonstrated that small transformers trained from scratch can learn unseen function classes in-context from examples.

In contrast to existing approaches, our study introduces a fresh perspective by viewing in-context learning in large language models (LLMs) as a unique machine learning algorithm. This conceptual framework enables us to leverage a classical tool from machine learning – analyzing decision boundaries in binary classification tasks. By visualizing these decision boundaries, both in linear and non-linear contexts, we gain invaluable insights into the performance and behavior of in-context learning. This method allows us to probe the inductive biases and generalization capabilities of LLMs and offers a unique assessment of the robustness of their in-context learning performance. Consequently, this approach provides a comprehensive means to qualitatively analyze the underlying mechanisms that govern in-context learning and suggest ways to improve its performance in LLMs.

To our surprise, we found that the recent LLMs struggle to provide smooth decision boundaries in all the classification tasks we considered, regardless of the model size, the number and ordering of in-context examples, and semantics of the label format. This issue persists even for simple binary linear classification tasks, where classical methods such as SVM can easily achieve smooth boundaries with fewer examples as shown in Figure 1. This observation raises questions about the factors that influence the decision boundaries of LLMs. To explore this, we experimented with a series of open-source LLMs including Llama2-7b, Llama2-13b, Llama3-8b [Touvron et al., 2023], Mistral-7b [Jiang et al., 2023], pruned Llama2-1.3b [Xia et al., 2023], as well as state-of-the-art closed-source LLMs GPT-4o and GPT-3-Turbo [Brown et al., 2020]. We then explore methods to smooth the decision boundary, including fine-tuning and adaptive prompting strategies. Our work provides valuable practical insights for understanding and improving in-context learning in LLMs through a new perspective. Our contributions can be summarized as follows:

- We introduce a novel mechanism to probe and understand in-context learning in LLMs by visualizing and analyzing the decision boundaries on classification tasks.
- We demonstrate that state-of-the-art LLMs exhibit non-smooth, irregular decision boundaries even on simple linearly separable tasks, unlike classical ML models.
- We study the influence of various factors impacting decision boundary smoothness, including model size, pretraining data and objectives, number of in-context examples, quantization levels, label semantics, and order of examples.
- We identify methods to improve the smoothness of LLM decision boundaries, such as fine-tuning earlier layers, fine-tuning on synthetic tasks and uncertainty-aware active learning.

## 2 Background

### 2.1 Training Large Language Models

Large Language Models (LLMs) are trained on vast corpora of text using unsupervised learning. During training, these models learn to predict the next token in a sequence. Given a sequence

of tokens $(x_1, x_2, \ldots, x_{t-1})$, the model predicts the next token $x_t$ by maximizing the likelihood $P(x_t|x_1, x_2, \ldots, x_{t-1})$. The training objective typically involves minimizing the cross-entropy loss:

$$L = -\sum_{i=1}^{N} \sum_{t=1}^{T_i} \log P(x_t|x_1, x_2, \ldots, x_{t-1}) \tag{1}$$

where $T_i$ is the number of tokens in the $i$-th sequence and $N$ is the total number of sequences in the corpus. During training, teacher forcing is often employed, where the model receives the ground truth token $x_t$ as input at each time step instead of its own prediction, enabling parallel training.

## 2.2 In-Context Learning in LLMs

After training, LLMs can generalize to new tasks through a mechanism known as in-context learning. Let $\mathcal{S} = \{(\mathbf{x}_1, y_1), (\mathbf{x}_2, y_2), \ldots, (\mathbf{x}_n, y_n)\}$ represent the set of $n$ input-output pairs provided as examples in the prompt, where $\mathbf{x}_i$ is an input and $y_i$ is the corresponding output. Given a new input $\mathbf{x}_{\text{new}}$, the LLM is turned into a task-specific model that predicts the output $\hat{y}_{\text{new}}$ by conditioning on the given examples: $P(\hat{y}_{\text{new}}|\mathbf{x}_{\text{new}}, \{(\mathbf{x}_1, y_1), (\mathbf{x}_2, y_2), \ldots, (\mathbf{x}_n, y_n)\})$. In-context learning allows the LLM to perform tasks by leveraging the context provided by these examples, thereby inferring the task and generating appropriate responses for new inputs. This approach utilizes the model's ability to recognize patterns and apply learned knowledge without additional training or fine-tuning.

## 3 Methodology

We aim to better understand in-context learning in Large Language Models by investigating their decision boundaries on a series of binary classification tasks. To increase the generality of our framework, we evaluate several existing LLMs on different task distributions under different settings. We present the general framework here, and refer to Section 4 for specific experiment settings.

## 3.1 In-Context Classification

Consider a $K$-class classification task with a data distribution $p_{\text{data}}(\mathbf{x}, y)$, where $\mathbf{x}$ is the input feature and $y \in \{1, \ldots, K\}$ is the class label. To construct an in-context prompt, we sample $n$ examples $(\mathbf{x}_i, y_i) \sim p_{\text{data}}$ for $i = 1, \ldots, n$. Given a new test point $\mathbf{x}_{\text{test}}$, in-context learning constructs a prompt $P = (\mathbf{x}_1, y_1, \ldots, \mathbf{x}_n, y_n, \mathbf{x}_{\text{test}})$ by concatenating the $n$ sampled examples and the test point. The prompt $P$ is then fed to the LLM $\pi$, which predicts a class $\hat{y}$ for $\mathbf{x}_{\text{test}}$.

We prompt the LLM with $P$ and obtain its prediction for $\mathbf{x}_{\text{test}}$ by choosing the most likely class in the next token distribution. Formally, let $V$ denote the size of the LLM's vocabulary, and $\mathbf{l} \in \mathbb{R}^V$ be the vector of logit values for each of the tokens. To obtain a class prediction, we convert each class label $i$ into a unique token id, say $c(i)$ and choose the class with the maximum logit value as the predicted label for $\mathbf{x}_{\text{query}}$, i.e., $\hat{y} = \arg\max_{i \in \{1, \ldots, K\}} l_{c(i)}$.

## 3.2 Decision Boundary Visualization

To visualize the decision boundary of a model $\pi$, we generate a grid of points covering the feature space defined by the in-context examples set $\mathcal{S}$. Let $\mathcal{S} = \{(\mathbf{x}_1, y_1), (\mathbf{x}_2, y_2), \ldots, (\mathbf{x}_k, y_k)\}$ represent the set of in-context examples, and $\mathbf{x}_{\text{min}}, \mathbf{x}_{\text{max}} \in \mathbb{R}^d$ denote the minimum and maximum values of the features in $\mathcal{S}$ along each dimension. We create a uniform grid with $G$ points along each dimension, resulting in a total of $G^d$ grid points. The grid points are denoted as $\mathbf{X}_{\text{grid}} = \{\mathbf{x}_{\text{query}} \mid \mathbf{x}_{\text{query}} \in [\mathbf{x}_{\text{min}}, \mathbf{x}_{\text{max}}]^d, \mathbf{x}_{\text{query}} = \mathbf{x}_{\text{min}} + i\Delta\mathbf{x}, i \in \{0, 1, \ldots, G-1\}\}$ where $\Delta\mathbf{x} = \frac{1}{G-1}(\mathbf{x}_{\text{max}} - \mathbf{x}_{\text{min}})$ is the grid spacing along each dimension. Each point $\mathbf{x}_{\text{query}} \in \mathbf{X}_{\text{grid}}$ is a query input, and the model $\pi$ is prompted with the sequence $(\mathbf{x}_1, y_1, \ldots, \mathbf{x}_k, y_k, \mathbf{x}_{\text{query}})$ to predict the corresponding class label $\hat{y}$. The decision boundary is then visualized by plotting the predicted labels $\hat{y}$ over the grid $\mathbf{X}_{\text{grid}}$.

# 4 Experiments

In this section, we examine existing LLMs through the lens of decision boundaries by conducting a series of binary classification tasks under varying conditions. Our experiments aim to address the following key questions:

- How do existing pretrained LLMs perform on binary classification tasks? §4.1

- How do different factors influence the decision boundaries of these models? §4.2

- How can we improve the smoothness of decision boundaries? §4.3

**Classification Tasks**. We investigate the decision boundary of LLMs by prompting them with $n$ in-context examples of binary classification tasks, with an equal number of examples for each class. We generate classification datasets using `scikit-learn` [Pedregosa et al., 2011], creating three types of linear and non-linear classification tasks: linear, circle, and moon, each describing different shapes of ground-truth decision boundaries. Detailed information on the dataset generation can be found in Appendix G. In addition to the in-context examples, we calculate the in-context learning accuracy on a held-out test set of size 100. We sample in-context examples and test points from classification task and convert them into prompt, with an example shown in Appendix F.

**Obtaining Decision Boundaries of Language Models**. We study an extensive range of models, with sizes ranging from 1.3B to 13B parameters, including open-source models such as Llama2-7B, Llama3-8B, Llama2-13B, Mistral-7B-v0.1, and sheared-Llama-1.3B. We also extend our analysis to state-of-the-art closed-source LLMs, including GPT-4o and GPT-3.5-turbo. We generate the decision boundaries of the open-source models with 8-bit quantization due to computational constraints. We choose a grid size scale of 50 x 50, resulting in 2500 queries for each decision boundary. For the open-source models, we use the approach described in 3.2 to get predictions. For the closed-source models, we use the next token generation as the prediction.

## 4.1 Non-Smooth Decision Boundaries of LLMs.

Figure 2 compares the decision boundaries of 6 LLMs when provided with 128 in-context examples. Even on simple linearly separable classification problems, all of these models exhibit non-smooth decision boundaries. The decision boundaries vary significantly across models, indicating that these models have different reasoning abilities to interpret the same in-context data. All models show fragmented decision regions, which means small changes in the input features can result in different classifications. This raises concerns about the reliability of LLMs and their practical deployment, as even when test accuracy for classification is high (shown in Figure 3, where test accuracy increases with the number of context examples), the underlying decision boundary lacks generalization. We further demonstrate nonsmoothness in NLP text classification tasks by projecting text input into 2D space, as detailed in Appendix H. In the following sections, we will explore factors that affect decision boundary smoothness and investigate methods to improve smoothness.

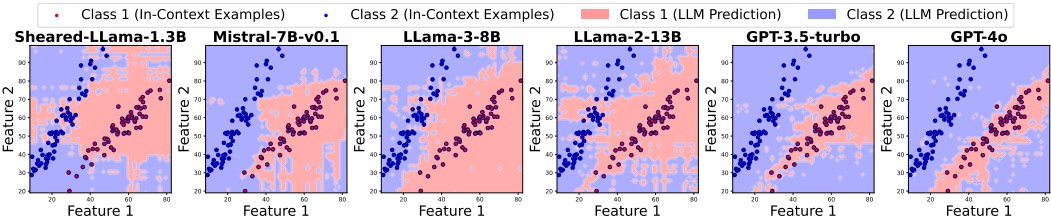

Figure 2: Visualizations of decision boundaries for various LLMs, ranging in size from 1.3B to 13B, on a linearly seperable binary classification task. The in-context data points are shown as scatter points and the colors indicate the label determined by each model. These decision boundaries are obtained using 128 in-context examples. The visualization highlights that the decision boundaries of these language models are not smooth.

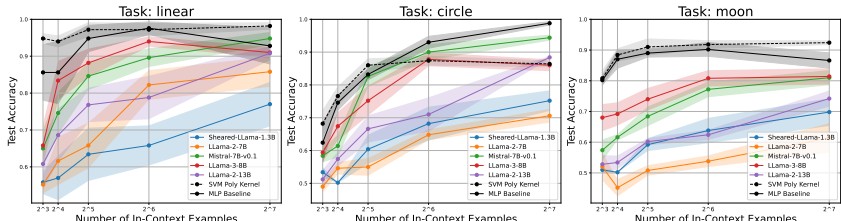

Figure 3: Test accuracy for LLMs and baselines across three classification tasks (linear, circle, and moon), with each subplot illustrating the test accuracy as the number of in-context examples increases. The baselines are the SVM with a polynomial kernel and the MLP with two hidden layers. Shaded regions represent the standard error of the mean accuracy across 5 seeds.

## 4.2 How Do Different Factors Influence the Decision Boundaries?

**Impact of Model Size on Decision Boundary and Accuracy** From Figure 2, model sizes increase from left to right, yet there is no clear correlation between model size and the smoothness of the decision boundary. Even the most powerful model, GPT-4o, demonstrates fragmented decision regions. This suggests that increasing model size alone is insufficient for improving decision boundary smoothness. However, as shown in Figure 3, larger models tend to perform better in terms of test accuracy compared to smaller models, with Llama-1.3B often performing the worst.

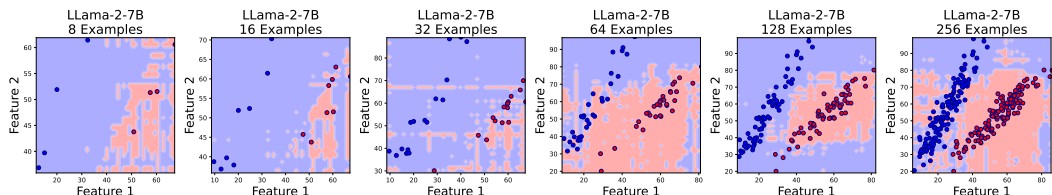

Figure 4: Decision boundary of Llama2-7b with increasing in-context examples from 8 to 256.

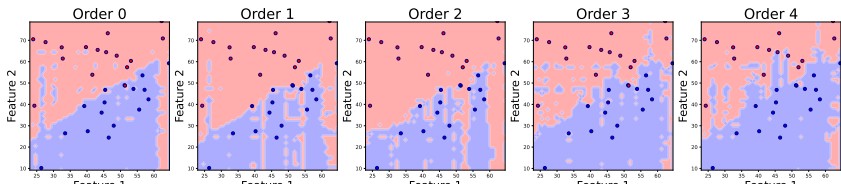

Figure 5: The sensitivity of the Llama3-8b model's decision boundary to the order of in-context examples. Each subplot (Order 0 to Order 4) shows the model's decision boundary with the same 32 examples shuffled differently.

**Increasing In-Context Examples Does Not Guarantee Smoother Decision Boundaries** While classification accuracies tend to improve with more in-context examples—and it's worth noting that Llama-3-8B and Mistral-7B's accuracy scales similarly to the SVM and MLP baselines—Figure 4 reveals that this does not translate to smoother decision boundaries. Despite the increase in accuracy, the decision boundaries remain fragmented, indicating that merely providing more in-context examples is not sufficient for achieving smoother decision regions.

**How Does Quantization Influence Decision Boundaries?** Figure 6a illustrates the decision boundaries of the LLaMA-2-7B model under different quantization levels [Dettmers et al., 2022]. When transitioning from 8-bit to 4-bit quantization, the red regions around the red in-context learning examples turn blue. This indicates that the reduced precision from 4-bit quantization significantly affects points near the decision boundary or areas where the model is most uncertain. For further investigation, we plot the probability prediction for class 1 (Figure 6b). The white regions, indicating a 50% probability for both classes, highlight the areas most impacted by quantization. Hence, varying quantization levels can flip the LLM's decisions in the regions of highest uncertainty.

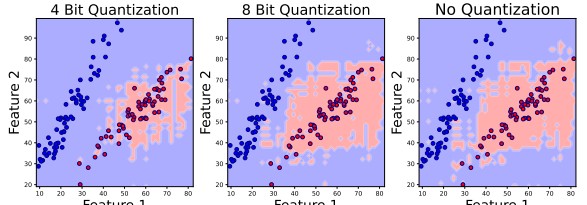 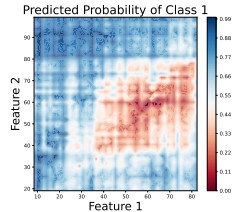

(a) Decision boundaries of Llama-2-7b with different quantization choices on a linearly seperable tsak.

(b) Prediction of probability of class 1 with 8-bit quantization.

Figure 6: Impact of quantization on Llama2-7-8b's decision boundaries and probability predictions.

**Are Decision Boundaries Sensitive to the Prompt Format?** Yes, decision boundaries are sensitive to the labels' names, as shown in Figure 7. Using semantically unrelated labels, such as "Foo" and "Bar" as suggested in [Wei et al., 2023], results in flipped predictions compared to using reversed class names like "Bar" and "Foo". This suggests that the LLM's prediction still depend on its semantic prior knowledge of the labels.

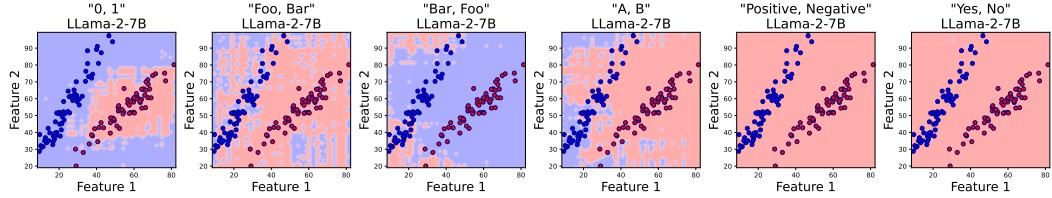

Figure 7: The decision boundaries of LLama-2-7B and LLama-3-8B, across various class labels. Each row corresponds to a model, and each column represents a different class label, shown in quotes.

**Are Decision Boundaries Sensitive to the Order of In-Context Learning Examples?** Recent works have shown that LLMs are sensitive to the order of in-context examples [Chen et al., 2024], which can significantly influence downstream performance. Similarly, as illustrated in Figure 5, we demonstrate that the model's decision boundaries vary with different shuffles of the in-context examples, highlighting the sensitivity of the decision boundaries to the order of the examples.

### 4.3 How to Improve the Decision Boundary Smoothness?

**Can We Finetune LLMs on the In-Context Examples to Achieve Smoother Decision Boundaries?** Our experiments indicate that finetuning LLMs on in-context examples does not result in smoother decision boundaries. Specifically, we finetuned Llama3-8B on 128 in-context learning examples and found that the resulting decision boundaries remained non-smooth. Examples of the decision boundaries after finetuning are provided in Appendix B.

**Can We Finetune LLMs on a Dataset of Classification Tasks to Achieve Smoother Decision Boundaries?** Previous works have shown that finetuning a pretrained LLM on a large collection of tasks improves its in-context learning performance on unseen tasks [Min et al., 2022a]. In this section, we investigate if the same paradigm helps improve the decision boundary smoothness of LLMs. To do this, we finetune a pretrained Llama model [Touvron et al., 2023] on a set of 1000 binary classification tasks generated from `scikit-learn` [Pedregosa et al., 2011], where the ground-truth decision boundary is either linear, circle-shaped, or moon-shaped, with equal probabilities. For each task, we sample randomly $N = 256$ data points $x \sim \mathbf{X}_{\mathrm{grid}}$ and their corresponding label $y's$. We then sample the number of context points $m \sim \mathcal{U}[8, 128]$, and finetune the LLM to predict $y_{i>m}$ given $x_{i>m}$ and the preceding examples:

$$\mathcal{L}(\pi) = \mathbb{E}\left[\sum_{i=m+1}^{N} \log p(y_i \mid x_i, x_{1:i-1}, y_{1:i-1})\right], \tag{2}$$

where the expectation is with respect to task, data points $\{(x_i, y_i)\}_{i=1}^{N}$, and the number of context points $m$. After training, we evaluate the same finetuned model on various binary classification tasks

with varying numbers of context points. To ensure the test tasks are unseen during training, we use different parameters in creating the datasets, such as the separateness between two classes and the scale between the inner and outer circles in the circle task. See Appendix G for more details.

We consider several finetuning settings for ablation studies. 1) In the first setting, we finetune the pretrained LLM using LoRA [Hu et al., 2021] and finetune the attention layers. 2) We finetune only the token embedding layer of LLM. 3) We finetune only the linear head layer of LLM. Then we consider modifying the architecture of the LLM: In this setting, we keep the core transformer backbone of the LLM frozen, attach randomly initialized embedding layers and prediction head to the model, and train the entire model using objective (2). This stems from the intuition that task-specific embedding and prediction layers allow the model to maximally utlize the general pattern-matching capabilities of the transformer backbone for the new task. We refer to this model as CustomLLM, and consider its three variants, which add 1) a new embedding layer for $x$, 2) a new prediction head for $y$, and 3) new embedding layers for $x$, $y$, and a new prediction head for $y$. The embedding layers and prediction head are MLPs with one hidden layer. We embed the raw numerical values instead of the text representation of $x$ whenever a new embeddding layer for $x$ is used (same for $y$), and predict directly the binary class values instead of text labels whenever the new prediction head is used. Results of Finetuning LLM and CustomLLM in Figure 8 and Figure 9 show that finetuning the intermediate and earlier embedding layers leads to smoother decision boundary compared to finetuning the top prediction head.

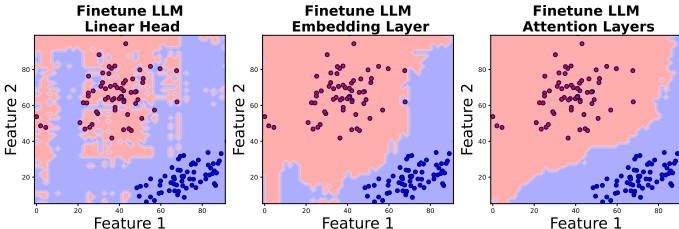

Figure 8: Decision boundary of Llama3-8B post finetuning the linear head, embedding layer and the attention layers. Finetuning the latter two layers improves the smoothness.

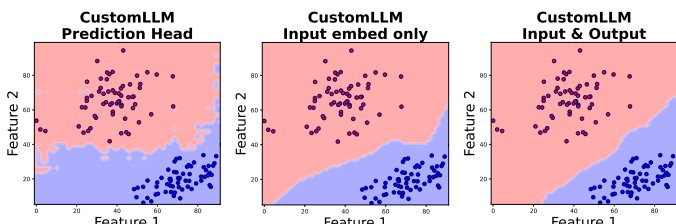

Figure 9: CustomLLM finetuning ablations. Decision boundary after finetuning the prediction head, input embedding layer and both layers for the CustomLLM.

**Can LLMs finetuned on one in-context learning task generalize to more complex in-context learning tasks?** In this section, we further explore whether a LLM fine-tuned only on a linear task can achiever smoother decision boundaries on unseen and more complex tasks. As shown in Figure 10, we compare the decision boundaries of Llama3-8b before and after SFT on the linear task only. Unexpectedly, we found it generalizes to unseen non-linear tasks as well as 3-class and 4-class classification tasks, despite only being trained on a binary linear task. The smoother decision boundaries observed in these unseen tasks suggest that fine-tuning on a synthetic in-context learning task can have downstream benefits for other tasks, enabling the model to be more robust in in-context learning.

**Can we train a transformer from scratch to learn smooth decision boundary in-context?** One may wonder whether a small transformer trained from scratch can provide smooth decision boundaries. To answer this, we train TNPs [Nguyen and Grover, 2022] , a transformer-based model specifically designed for in-context learning. For each sequence of data points $\{(x_i, y_i)\}_{i=1}^{N}$ from a task $C$, TNPs learn to predict the query labels $y_{i>m}$ given the query inputs $x_{i>m}$ and the context pairs, assuming

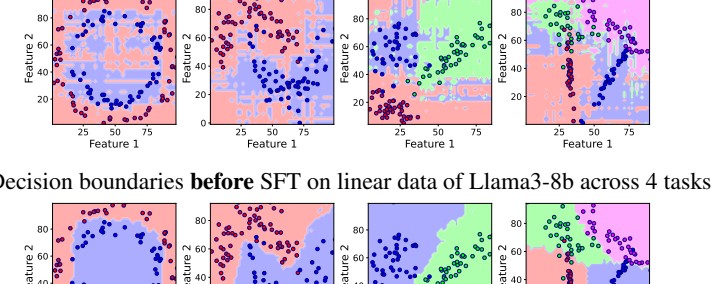

(a) Decision boundaries **before** SFT on linear data of Llama3-8b across 4 tasks.

(b) Decision boundaries **after** SFT on linear data of Llama3-8b across 4 unseen tasks.

Figure 10: Generalization ability of Llama-3-8B after supervised fine-tuning on a single binary linear classification task. The first two columns show the model's performance on non-linear classification tasks before and after fine-tuning, while the last two columns demonstrate its ability to generalize to 3-class and 4-class classification tasks.

conditional independence among the queries given the context:

$$\mathcal{L}(\theta) = \mathbb{E}\left[\sum_{i=m+1}^{N} \log p(y_i \mid x_i, x_{1:m}, y_{1:m})\right], \tag{3}$$

where the expectation is with respect to task $C$, data points $\{(x_i, y_i)\}_{i=1}^{N}$, and the number of context points $m$. TNPs employ a specialized mask to ensure the conditional independence assumption. We showed in Appendix D that transformers trained from sctrach can learn to in-context learn smooth decision boundary. Details are in Appendix D.

**How to Use Uncertainty-aware Active Learning to Learn Decision Boundaries** We investigate whether the decision boundary can be smoothed by providing the LLM with labels of the most uncertain points on the grid as additional in-context examples. Uncertainty is measured as the entropy of the probability distribution of the two classes after softmax normalization of the logits. Our study focuses on an active learning scheme where new in-context examples are incrementally added based on the LLM's current uncertainty. Initially, we obtain the decision boundary conditioned on the existing in-context examples. To refine this boundary, we query the LLM over a grid and select the top-k most uncertain points, ensuring they are spatially distant from each other using a greedy sampling approach. For labeling these uncertain points, we use a logistic regression model well-trained on a larger dataset with perfect accuracy as the ground truth decision boundary. As shown in Figure 11, this uncertainty-aware active sampling method results in a smoother decision boundary over iterations compared to random sampling. The iterative refinement enhances the model's generalization capabilities, leading to higher test set accuracies and greater sample efficiency, requiring fewer additional in-context examples to achieve performance gains. These findings indicate that leveraging the LLM's uncertainty measurements is valuable for selecting new in-context examples in resource-constrained settings where labeled data is scarce. We show more examples in Appendix I.

## 5   Related Works

Understanding in-context learning in transformers and LLMs is an active area of research, with existing works approaching this problem from both theoretical and practical perspectives.

**Theoretical understanding of in-context learning** Recent works aim to establish a theoretical connection between in-context learning and gradient descent (GD). The pioneering work by Akyürek et al. proves transformers can implement learning algorithms for linear models based on GD and closed-form ridge regression by construction. Von Oswald et al. [2023] proves the equivalence between linear self-attention and GD on linear regression by construction. Similarly, Dai et al.

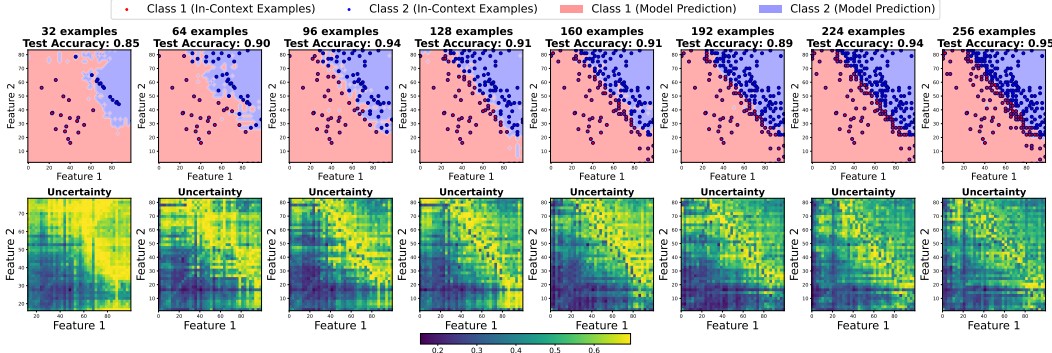

(a) Decision boundaries with different numbers of context examples when using active sampling.

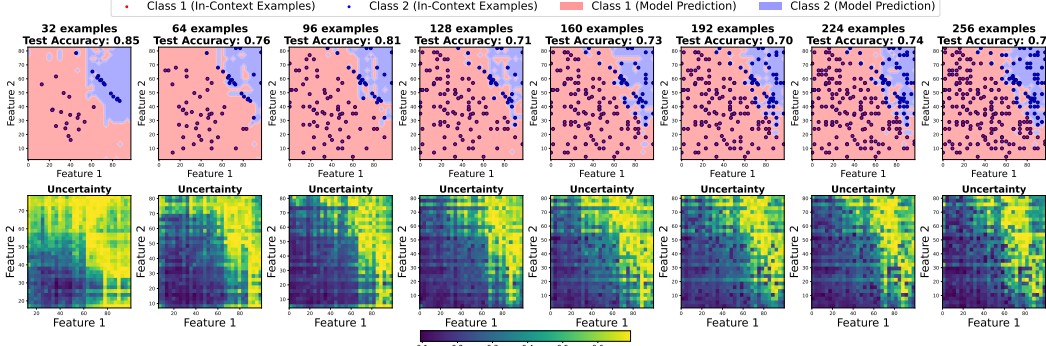

(b) Decision boundaries with different numbers of context examples when using random sampling.

Figure 11: Comparison of active and random sampling methods. We plot the decision boundaries and uncertainty plot across different number of in-context examples from 32 to 256, where the in-context examples are gradually added to the prompt using active or random methods. Active sampling gives smoother decision boundary and the uncertain points lie on it. The test set accuracies is plotted in the titles.

[2023] shows that attention in transformers has a dual form of GD and views transformers as meta-optimizers. Subsequent works extend these ideas to characterize the global optimum of single-layer linear transformers. Ahn et al. [2024] observe that with the optimal parameters, the transformer implements a single step of preconditioned gradient descent, while Zhang et al. [2023] shows that at the global optimum, the transformer achieves competitive prediction error with the best linear predictor on a new prediction task. In addition to theoretical connections to GD, a complementary direction aims to establish statistical complexity and generalization bounds of in-context learning in transformers [Bai et al., 2024, Li et al., 2023b, Wies et al., 2024, Wu et al., 2023]. The common limitation of these existing theoretical frameworks is the reliance on strong assumptions about the transformer architecture or the functional form of the in-context learning tasks which may not reflect real-world practices.

**Practical understanding of in-context learning** More relevant to our paper is a line of works focusing on understanding the practical aspects of in-context learning in LLMs. Many existing works investigate the roles of in-context examples and prompts. Min et al. [2022b] show a surprising result that ground-truth demonstrations are not required for in-context learning, while other factors such as the label space, input text distribution, and overall sequence format play an important role. Shi et al. [2023] investigate the distractibility of LLMs and shows that their performance dramatically drops when irrelevant context is included. Subsequently, Wei et al. [2023] characterize these behaviors of LLMs with respect to model size, and show that larger language models perform in-context learning differently in the presence of flipped or semantically unrelated labels. Webson and Pavlick [2022] argue against the current practice of prompt engineering, showing that intentionally irrelevant or even pathologically misleading prompts achieve similar downstream performance to instructively good prompts. Orthogonally, Lampinen et al. [2022] find that including explanations in the in-context

examples significantly improves the few-shot performance of LLMs. Finally, given the expanded context windows of modern LLMs, recent works have explored in-context learning in the many-shot setting with hundreds or thousands of examples [Agarwal et al., 2024, Li et al., 2023a, Bertsch et al., 2024].

**Learning to learn in-context** In contrast to the emergent in-context capabilities of LLMs, existing works have also studied methods that learn to perform in-context learning explicitly. Min et al. [2022a] propose MetaICL, a meta-training framework for finetuning pretrained LLMs to perform in-context learning on a large and diverse collection of tasks. MetaICL outperforms several baselines including emergent in-context learning and multi-task learning followed by zero-shot transfer. Beyond text, TNP [Nguyen and Grover, 2022, Nguyen et al., 2023, Nguyen and Grover, 2024] and PFNs [Müller et al., 2021] propose to train transformer models to perform in-context prediction for a family of functions, which allows in-context generalization to unseen functions after training. Similarly, Garg et al. [2022] show that autoregressive transformers can be trained from scratch to learn function classes such as linear functions and 2-layer ReLU networks. Other work also shows that alignment can be done in-context [Zhao et al., 2023], where in-context learned reward model can be used for inference-time preference alignment. These works present an interesting set of baselines for our work to examine the in-context learning ability of LLMs.

# 6 Conclusion

We propose a novel approach to understanding in-context learning in LLMs by probing their decision boundaries in in-context learning in binary classification tasks. Despite achieving high test accuracy, we observe that the decision boundaries of LLMs are often irregularly non-smooth. Through extensive experiments, we identify factors that affect this decision boundary. We also explore fine-tuning and adaptive sampling methods, finding them effective in improving boundary smoothness. Our findings provide new insights into the mechanics of in-context learning and suggest pathways for further research and optimization.

# Acknowledgments

This research is supported by NSF CAREER Award 2341040, Schmidt Sciences AI2050 Fellowship, Samsung, and Cisco.

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

# A Pretrained LLMs decision boundary on linear and non-linear classification tasks

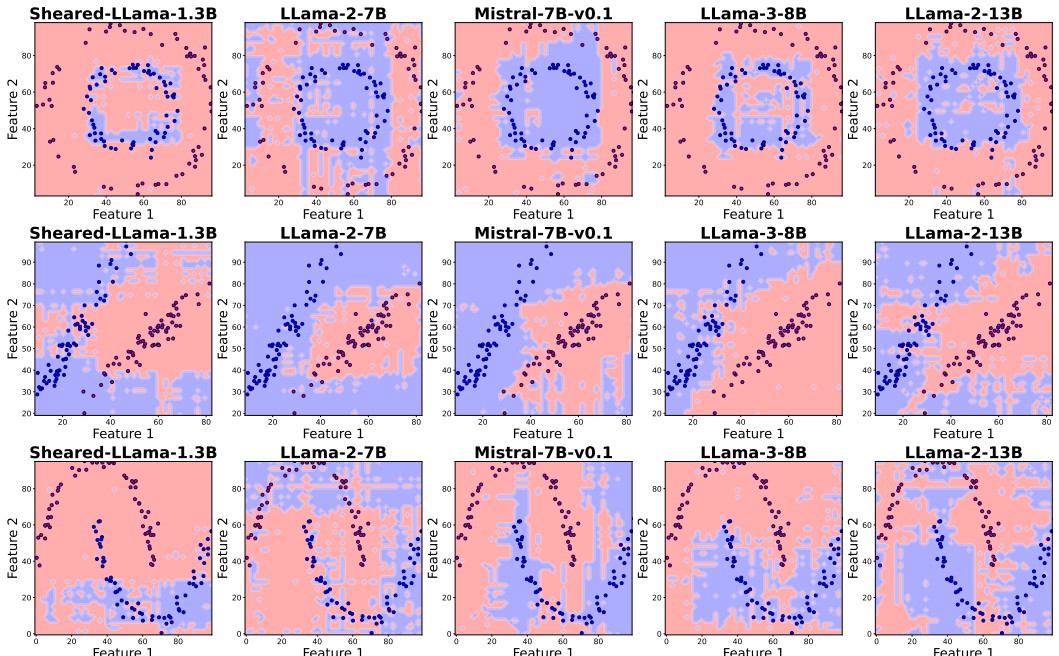

Figure 12: Visualizations of decision boundaries for various LLMs, ranging in size from 1.3B to 13B, on three classification tasks. The tasks are, from top to bottom, circle, linear, and moon classifications. Note that the circle and moon tasks are not linearly separable. The in-context data points are shown as scatter points and the colors indicate the label determined by each model. These decision boundaries are obtained using 128 in-context examples. The visualization highlights that the decision boundaries of these language models are not smooth.

# B Finetune on in-context examples only

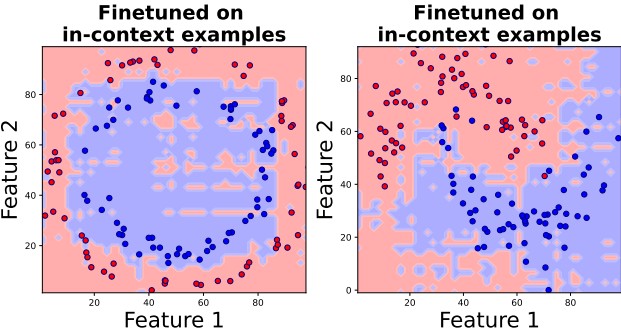

Figure 13: Two examples of Llama2-7B finetuned on the in-context examples points, which are scattered points in the plot.

# C  SFT LLMs for in-context classification

We used LoRA [Hu et al., 2021] to supervise fine-tune the Llama series models on both non-linear and linear classification tasks, including circle, linear, and moon datasets. The models fine-tuned are Sheared-Llama-1.3B, Llama2-7B, Llama2-13B, and Llama3-8B. Visualization in Figure 14 demonstrates that these language models produce smoother decision boundaries after training on the classification datasets using SFT.

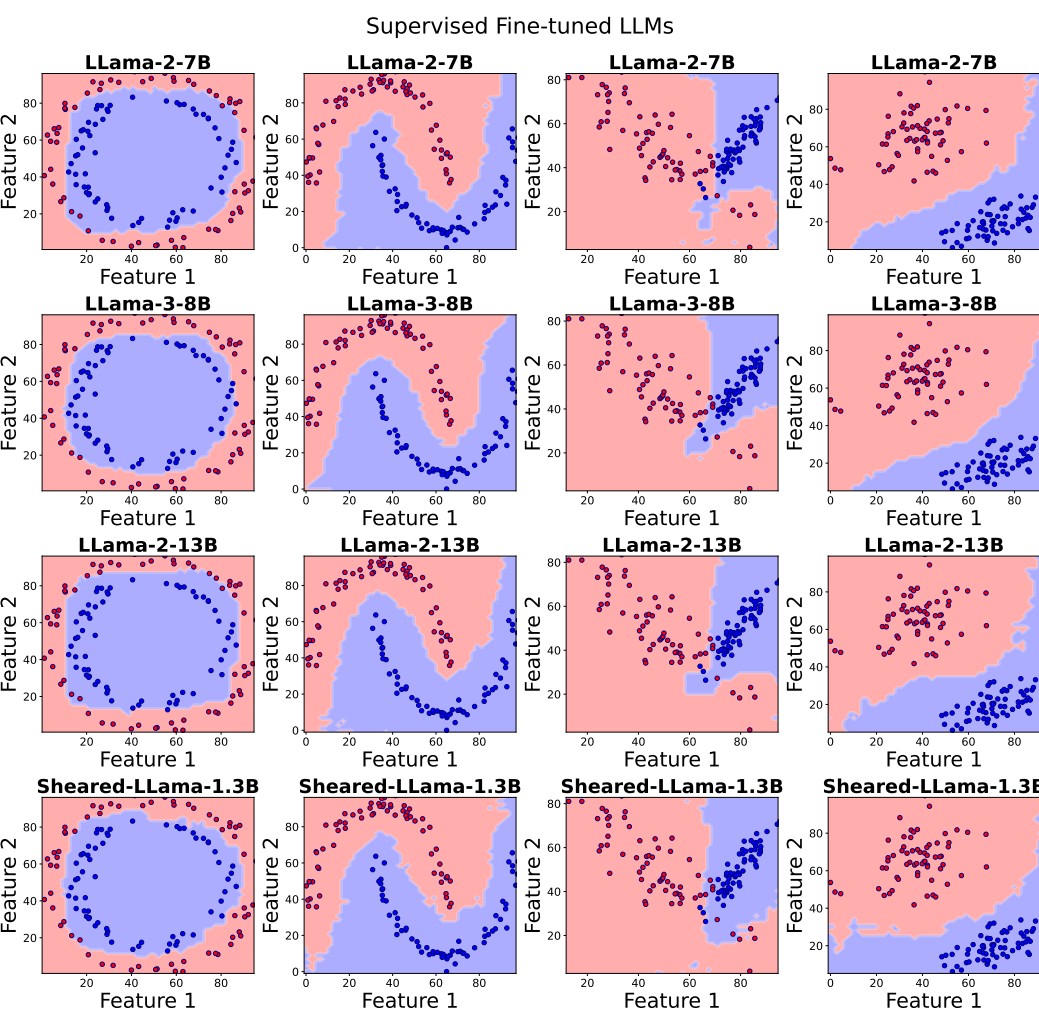

Figure 14: Decision boundary of in-context learning on 128 examples across Llama series models after supervised finetuning with LoRA.

# D   Training Transformers from Scratch: TNP models decision boundaries

We trained TNP models of four different sizes as shown in the Table 1 below. We plot how does the TNP models decision boudnary changes as more in-context examples are added in Figure 15. TNP models learn smooth deicision boundary for this moon-shaped non-linear task. And we did not observe a scaling law of transformer sizes versus the decision boundary smoothness. In contrast the smaller model generalize better than the larger model.

Table 1: TNP transformers model sizes and architectures.

| Model | Parameters (M) | Input embed dim | feedforward dim | num heads | num layers |
|---|---|---|---|---|---|
| Small | 0.1 | 64 | 64 | 2 | 3 |
| Medium | 0.6 | 128 | 128 | 4 | 6 |
| Large | 1.6 | 128 | 256 | 8 | 12 |
| X-Large | 9.7 | 256 | 512 | 16 | 18 |

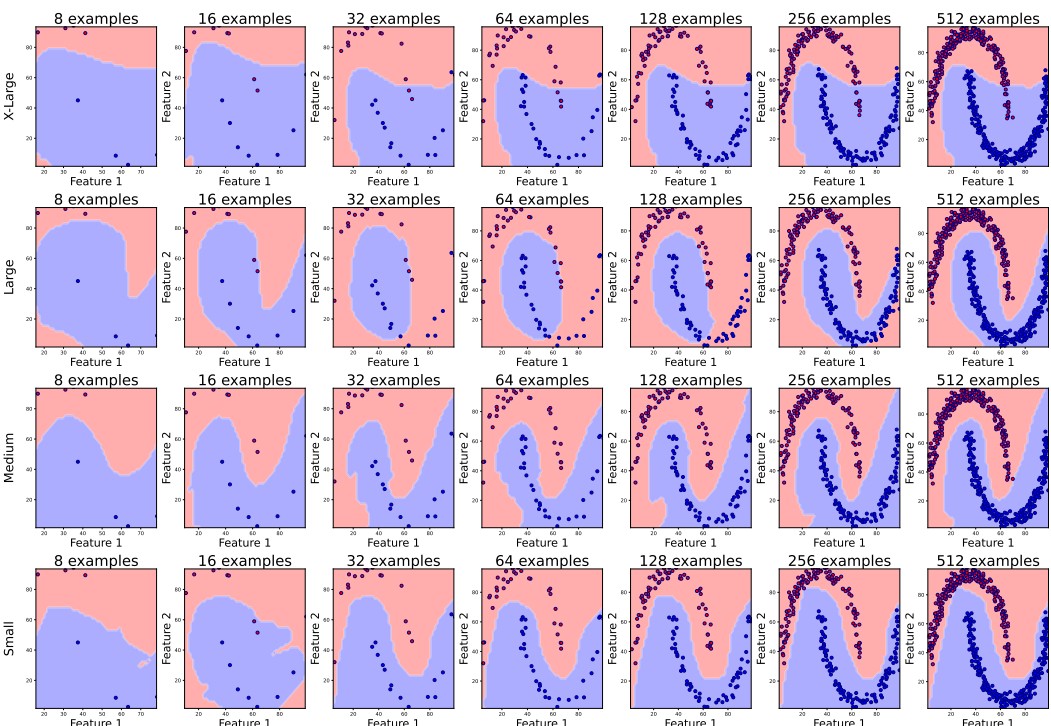

Figure 15: Decision boundary of TNP models of different sizes trained from scratch.

# E   Traditional Classifiers Model Details

In our experiments, we used several classical machine learning models with the following hyperparameters:

- **Decision Tree Classifier:** We set the maximum depth of the tree to 3.

- **Multi-Layer Perceptron:** The neural network consists of two hidden layers, each with 256 neurons, and the maximum number of iterations is set to 1000.

- **K-Nearest Neighbors:** The number of neighbors is set to 5.

- **Support Vector Machine (SVM):** We used a radial basis function (RBF) kernel with a gamma value of 0.2.

## F    Prompt Format for binary classification

```
Given pairs of numbers and their labels, predict the label for a new
input pair of numbers based on the provided data.
Answer with only one of the labels 'Foo' and 'Bar':

Input:  64 24
Label:  Bar
Input:  34 41
Label:  Bar
Input:  71 66
Label:  Bar
...
Input:  96 49
Label:  Foo
Input:  21 56
Label:  Foo

What is the label for this input?
Input:  2 3
Label:
```

Figure 16: Few-shot in-context prompt with $n$ context questions.

## G    Classification Datasets

We use three types of classification tasks from `scikit-learn` [Pedregosa et al., 2011] to probe the decision boundary of LLMs and transformers: linear, circle, and moon classification problems. For linear classification tasks, we utilize the `make_classification` function, which generates random classification problems by creating clusters of points normally distributed around the vertices of a hypercube with sides of length $2 \times$ class_sep. Circle classification tasks are generated using the `make_circles` function, creating a binary classification problem with a large circle containing a smaller circle. The `factor` parameter controls the scale of the inner circle relative to the outer circle. Moon classification tasks are generated using the `make_moons` function, creating a binary classification problem with two interleaving half circles. The `noise` parameter controls the standard deviation of Gaussian noise added to the data points.

For training tasks, the `class_sep` parameter is randomly sampled from the range $[1.5, 2]$, and the `factor` parameter for circular tasks is sampled from $[0.1, 0.4]$. For testing tasks, the `class_sep` parameter is sampled from $[1, 1.4]$, and the `factor` parameter from $[0.5, 0.9]$, ensuring that testing tasks differ from training tasks. The `noise` parameter for moon-shaped tasks is sampled from $[0.05, 0.1]$ for training and $[0.1, 0.2]$ for testing, introducing varying levels of complexity in the classification problems.

## H    Decision Boundary of LLMs on NLP tasks.

We extend our analysis to multi-class NLP classification tasks using high-dimensional real-world datasets. To address the challenge of visualizing high-dimensional text embeddings, we project them onto a 2D space using t-SNE and send the 2D embeddings as input in the prompt to the LLM. While any dimensionality reduction technique inevitably introduces confounding factors, this approach allows us to extend our analysis to more complex, real-world scenarios. Our experiments encompass six widely-used NLP classification tasks, covering both binary and multi-class settings. These include Subjective/Obejective sentence classification (SUBJ) [Conneau and Kiela, 2018], financial sentiment analysis (FP) [Malo et al., 2014], textual entailment recognition (RTE) [Wang et al., 2019], hate speech detection (ETHOS) [Mollas et al., 2020], sentiment analysis (SST-2) [Socher et al., 2013]

and news topic classification (AG_NEWS) [Zhang et al., 2015]. The results, presented in Figure 17, demonstrate that the non-smooth decision boundary characteristics observed in our synthetic datasets persist in these more complex NLP tasks.

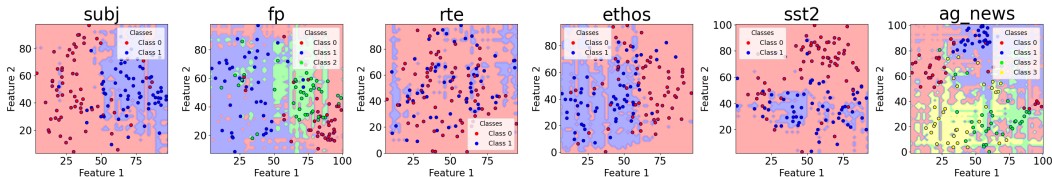

Figure 17: Decision boundaries of Llama-3-8b on six NLP tasks, ranging from binary to multi-class classification. Since text embeddings are natively high-dimensional, we projected text embeddings onto a 2D space using t-SNE. The irregular, non-smooth behaviors are also seen in these tasks.

# I   Uncertainty Aware Active Sampling For Smoother Decision Boundary and Better Test set Accuracy

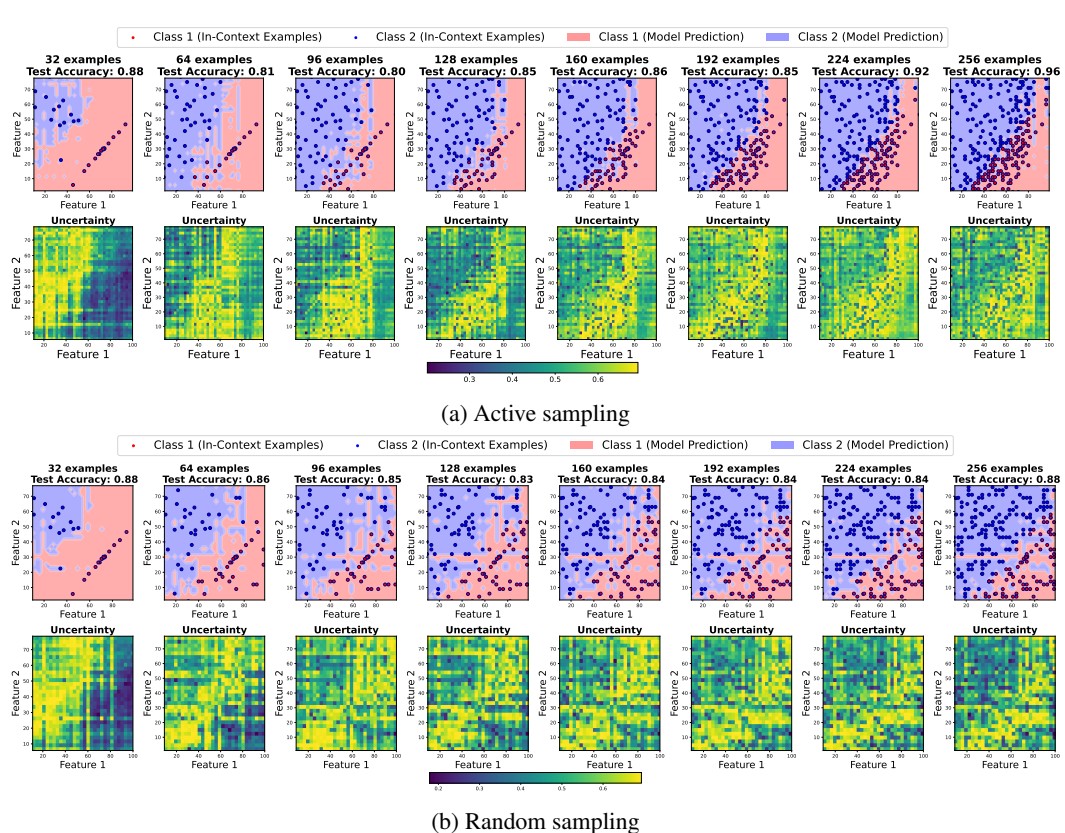

Figure 18: Comparison of active and random sampling methods.

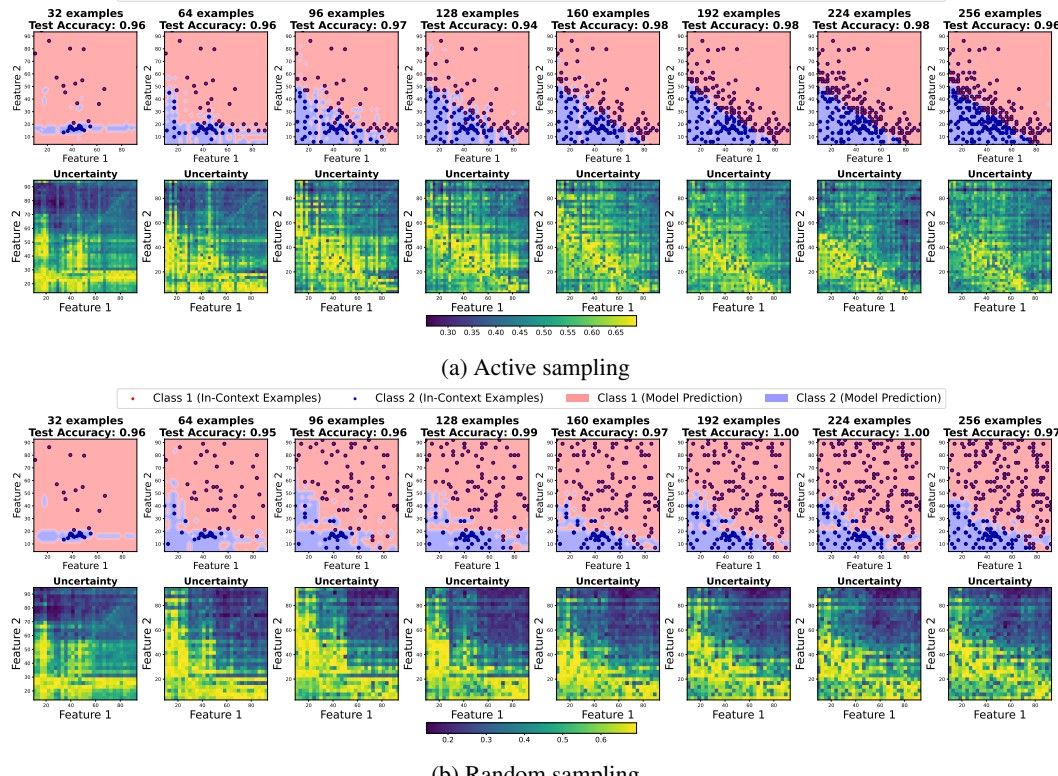

(a) Active sampling

(b) Random sampling

Figure 19: Comparison of active and random sampling methods.

