# OpenReview forum: "Probing the Decision Boundaries of In-context Learning in Large Language Models"
_NeurIPS.cc/2024/Conference — NeurIPS 2024 poster_

### Official Review · Reviewer_AvnH · 2024-07-12

**Soundness:** 3
**Presentation:** 3
**Contribution:** 3
**Rating:** 6
**Confidence:** 4

**Summary:**

This study investigate in-context learning in LLMs by examining their decision boundaries in binary classification tasks. The authors investigate the performance of several mainstream models on these tasks. Despite achieving high test accuracy, the decision boundaries of these LLMs are often irregularly non-smooth. Factors such as model size and prompts' influence on decision boundaries are explored. Additionally, the author examines fine-tuning and adaptive sampling methods, finding them effective in improving boundary smoothness.

**Strengths:**

- This is the first study, to my knowledge, to explore the decision boundaries of in-context learning LLMs.
- The experiments are thorough, and some findings, such as the use of uncertainty-aware active learning to help LLMs learn decision boundaries, are beneficial for future research.

**Weaknesses:**

- The tests are mainly conducted on binary classification tasks, making it unclear if the findings can be generalized to other tasks.
- Although different model sizes are covered, these models are from different series. The reason for using these models should be clarified in the paper.
- The decision boundaries appear to be heavily influenced by the prompts, such as the design of labels and the order of in-context learning examples. Coule the authors explore the robustness of their results to variations in prompts, like different formats or synonym replacements? If the robustness to prompts is poor, applying these results to other fields could be challenging.

**Questions:**

- Line 69: “Large Language Models (LLMs) are trained on vast corpora of text using unsupervised learning.” Shouldn't this be self-supervised learning?

- Line 132-134: “For the open-source models, we use the approach described in 3.2 to get predictions. For the closed-source models, we use the next token generation as the prediction…” This is confusing since section 3.2 only mentions visualization methods. What are the differences in prompt design or experiments between these two approaches?

**Limitations:**

Overall, this is a good paper. I would raise my score if the authors could address my main concerns, especially regarding the robustness of their results to variations in prompts.

---

> ### Author Response · Authors · 2024-08-07
>
> Dear Reviewer AvnH,
>
> Thank you for your feedback. We hope to address your questions and comments below.
>
> > Q1:"The tests are mainly conducted on binary classification tasks, making it unclear if the findings can be generalized to other tasks."
> 1. Our primary motivation for using synthetic 2D classification datasets is their amenability to decision boundary visualization, which is central to our paper's focus as decision boundaries are typically studied in 2D space, and our proposed mechanism offers novel insights into the generalization abilities of in-context learning. NLP classification tasks, with their discrete token nature and variable input lengths, pose significant challenges for visualization of decision boundaries along x and y axes.
> 2. Our use of synthetic function datasets is firmly grounded in established practices for studying in-context learning. Several recent high-profile works have exclusively used synthetic data to investigate ICL/learning algos:
>     - NeurIPS 2022 paper (https://arxiv.org/pdf/2208.01066) focuses entirely on linear functions for ICL.
>     - ICLR 2023 paper  (https://arxiv.org/pdf/2211.15661) explores learning algorithms for ICL and primarily conducts experiments on synthetic linear functions.
>     - Google's research (https://arxiv.org/abs/2303.03846) employs synthetic linear function classes (see Section 6).
>     - NeurIPS 2019 paper (https://arxiv.org/pdf/1905.11604) utilizes similar scikit-learn datasets to study decision boundaries for SDE algorithms (see Figure 1).
>
> These papers demonstrate that our approach is quite prevalent in the field.
>
> 3. To extend beyond binary tasks, we conduct new experiments on 6 real-world NLP classification datasets. Note, to handle the high dimensionality of text embeddings, we projected them onto 2D space using t-sne. (Any dimensionality reduction technique will introduce confounders to the analysis, but this price is inevitable to extend our analysis). We experiment with a total of 6 widely-used NLP classification tasks, including both binary and multi-class settings. These include sentiment analysis and Textual Entailment Recognition, providing a broader perspective on the applicability of our approach. Our results, presented in **Figure 1 of the rebuttal PDF**, demonstrate similar non-smooth decision boundary characteristics in NLP tasks as observed in our synthetic datasets.
>
> > Q2: "Although different model sizes are covered, these models are from different series. The reason for using these models should be clarified in the paper."
>
> We use these LLMs to provide a comprehensive analysis across different sizes and architectures, representing current SoTA open-source LLMs. For the sizes, due to computational constraints in an academic setting and the expensive nature of querying decision boundaries with large grid points, we limited our analysis to models no larger than 13B parameters. Therefore to complete the size series and gain insights into smaller-scale models, we included the pruned llama 1.3B model.
>
> > Q3: "The decision boundaries appear to be heavily influenced by the prompts, such as the design of labels and the order of in-context learning examples. ... explore the robustness of their results to variations in prompts, like different formats or synonym replacements?"
>
> To address this concern, we have conducted experiments to explore how decision boundaries are affected by prompt formats, using 4 different synonyms for key terms. As shown in **Figure 2 of the rebuttal PDF**, we find that the LLM's decision boundary is indeed affected by the prompt format, which aligns with the importance of prompt engineering in ICL. However, the overall non-smoothness level of the decision boundaries remained consistent. We view prompt formats as another factor influencing the decision boundary rather than undermining our results, since our central observations regarding the non-smoothness of decision boundaries show similar patterns across different prompts. We will add this as another influencing factor in our paper.
>
> > Q4: “Large Language Models are trained on vast corpora of text using unsupervised learning.” Shouldn't this be self-supervised learning?
>
> Sorry for the typo! We will revise this.
>
> > Q5: “For the open-source models, we use the approach described in 3.2 to get predictions. For the closed-source models, we use the next token generation as the prediction…” This is confusing since section 3.2 only mentions visualization methods. What are the differences in prompt design or experiments between these two approaches?
>
> We apologize for the confusion. For both open and closed source models, we used the same prompts. For open source models, we used logits to get predictions for the class labels. For closed source models, we looked at the predictions since the logits were not available to us.
>
> We hope this addresses your concerns. Please let us know if you have any questions.

---

> > ### Comment · Reviewer_AvnH · 2024-08-12
> > **Response to the author**
> >
> > Thank you. Considering this is the first study on decision boundaries in LLMs (as far as I know) and the experiments are solid, I have raised my score.

---

> > > ### Author Response · Authors · 2024-08-12
> > > **Thank you for raising score!**
> > >
> > > We thank the reviewer for raising the score and are pleased that you find our work novel and our experiments solid.

---

### Official Review · Reviewer_PktE · 2024-07-12

**Soundness:** 2
**Presentation:** 3
**Contribution:** 2
**Rating:** 3
**Confidence:** 3

**Summary:**

This paper investigates the decision boundary of in-context learning of Transformers. The paper shows that for three toy tasks, the decision boundaries of in-context learning of various pretrained models are not smooth. The paper then explores the method for improving the smoothness of decision boundaries and finds that supervised finetuning can mitigate this non-smoothness. Furthermore, finetuning on one task can benefit the smoothness of decision boundaries on other tasks.

**Strengths:**

1. The decision boundary of in-context learning is an interesting research topic, which seems not investigated by previous work.
2. The paper provides multiple empirical results and finds a method for improving the smoothness of decision boundary of in-context learning.
3. The writing of this paper is clear.

**Weaknesses:**

1. The empirical results are limited to toy datasets, which may reduce the impact of the proposed smoothness-improving method.
2. The paper does not empirically show that smoother decision boundary smoothness leads to higher generalization accuracy. On the contrary, the paper finds that gpt-4o has high accuracy along with not smooth decision boundary, which makes improving smoothness less motivated.
3. No quantitative evaluation of smoothness is present in the papers, making smoothness difficult to compare.

**Questions:**

See the weaknesses.

**Limitations:**

The main limitation is that the experiments are conducted only on toy datasets.

---

> ### Author Rebuttal · Authors · 2024-08-07
>
> Dear Reviewer PktE,
>
> Thank you for your feedback. We hope to address your questions and comments below.
>
> > "The empirical results are limited to toy datasets, which may reduce the impact of the proposed smoothness-improving method."
>
> 1. Our primary motivation for using synthetic 2D classification datasets is their amenability to decision boundary visualization, which is central to our paper's focus as decision boundaries are typically studied in 2D space, and our proposed mechanism offers novel insights into the generalization abilities of in-context learning. NLP classification tasks, with their discrete token nature and variable input lengths, pose significant challenges for visualization of decision boundaries along x and y axes.
> 2. Our use of synthetic function datasets is firmly grounded in established practices for studying in-context learning. Several recent high-profile works have exclusively used synthetic data to investigate ICL/learning algos:
>
>     - NeurIPS 2022 paper (https://arxiv.org/pdf/2208.01066) focuses entirely on linear functions for ICL.
>     - ICLR 2023 paper  (https://arxiv.org/pdf/2211.15661) explores learning algorithms for ICL and primarily conducts experiments on synthetic linear functions.
>     - Google's research (https://arxiv.org/abs/2303.03846) employs synthetic linear function classes (see Section 6).
>     - NeurIPS 2019 paper (https://arxiv.org/pdf/1905.11604) utilizes similar scikit-learn datasets to study decision boundaries for SDE algorithms (see Figure 1).
> These papers demonstrate that our approach is quite prevalent in the field. The use of scikit-learn is simply a means to generate widely accepted and reproducible benchmarks with clear characteristics such as linear/non-linear patterns.
>
> 3. Following up on the reviewers’ suggestions, we include additional experiments on 6 high-dimensional real-world classification datasets. Note, to handle the high dimensionality of text embeddings, we projected them onto 2D space using t-sne. (Any dimensionality reduction technique will introduce confounders to the analysis, but this price is inevitable to extend our analysis). We experiment with a total of 6 widely-used NLP classification tasks, including both binary and multi-class settings. These include sentiment analysis and Textual Entailment Recognition, providing a broader perspective on the applicability of our approach. Our results, presented in **Figure 1 of the rebuttal PDF**, demonstrate similar non-smooth decision boundary characteristics in NLP tasks as observed in our synthetic datasets.
>
> > "The paper does not empirically show that smoother decision boundary smoothness leads to higher generalization accuracy. On the contrary, the paper finds that gpt-4o has high accuracy along with not smooth decision boundary, which makes improving smoothness less motivated."
>
> - While we do not conclusively link boundary smoothness to NLP task performance, this was not our primary aim. Our research aims to provide an initial exploration of decision boundary characteristics in in-context learning, a crucial yet understudied area for understanding ICL behavior.
> - Following up on the reviewer’s suggestion, to investigate the benefits of smoother boundaries, we fine-tuned a Llama3 8B model on synthetic classification tasks and tested it on 7 diverse NLP datasets on ICL. As shown in the **Table 1 of the rebuttal pdf**, the results were promising: the fine-tuned model showed significantly higher performance on several tasks and improved average performance across seven tasks, suggesting smoother boundaries can contribute to better generalization in NLP tasks.
> - The fact that GPT-4o's has high accuracy and a less smooth decision boundary does not negate the importance of decision boundary smoothness. Decision boundary characteristics often imply generalization to unseen data. GPT-4's high accuracy on current benchmarks does not necessarily imply good generalization, as its training data is not fully known. Different models may achieve high performance through various mechanisms, and smoothness could be beneficial, particularly for specific tasks that require high reliability, robustness, and interpretability.
>
> > "No quantitative evaluation of smoothness is present in the papers, making smoothness difficult to compare."
>
> - Formally measuring smoothness is challenging due to its dependence on grid discretization. Qualitatively, the difference in smoothness is evident when comparing traditional models (e.g., kNN, decision trees, logistic regression) to LLMs. To address this concern quantitatively, we explored several metrics, including curvature (average rate of change of the tangent vector along the decision boundary) and boundary length (total length of the decision boundary). However, measuring the continuity behavior of decision boundaries with discretized evaluations is non-trivial and these quantitative metrics are challenging to define and often do not show obvious trends.
>
> - Nevertheless, we define and present an empirical metric for decision boundary smoothness: Nearest Neighbor Entropy. This measure calculates the average entropy of predictions among the k-nearest neighbors for each point in the grid, reflecting the variability and smoothness of the boundary.
>
> - We show in **Table 2 of the rebuttal PDF** the NN entropy values for the models shown in Figure 1 of our submitted paper, comparing the smoothness of traditional models and state-of-the-art LLMs. LLMs exhibit higher NN entropy than models like Decision Trees and kNN. We also provide results for different LLMs with 64 and 128 in-context examples, demonstrating that as the number of examples increases, the entropy generally decreases. This quantitative approach, combined with our qualitative observations, provides a more comprehensive assessment of decision boundary smoothness.
>
> We hope this addresses your concerns. Please let us know if you have any questions.

---

> ### Author Response · Authors · 2024-08-12
>
> Dear Reviewer PktE,
>
> Thank you again for your helpful feedback! Your suggestions have helped us solidify our experiments and analysis by:
>
> 1. Conducting additional experiments on NLP tasks spanning 6 NLP classification datasets,
> 2. Empirically demonstrating that smoothness on synthetic datasets can lead to improvements on downstream tasks, on average, across 7 ICL NLP tasks. These first two experiments address your main concern that "the main limitation is that the experiments are conducted only on toy datasets."
> 3. Using quantitative metrics, such as nearest neighbor entropy, as a measure of smoothness.
>
> We hope this addresses the concerns you listed in your review. As the discussion period is nearing its end, could you please let us know if you have any further questions for us? We are happy to address any questions or concerns you may have.
>
> Thank you for your time!

---

### Official Review · Reviewer_WmaA · 2024-07-13

**Soundness:** 4
**Presentation:** 4
**Contribution:** 4
**Rating:** 10
**Confidence:** 4

**Summary:**

The authors study the decision boundaries of LLMs in binary in-context learning tasks, finding that decision boundaries can be non-smooth despite high test accuracy and linear separability of the task itself. They examine numerous methods for smoothing the decision boundaries, including SFT and training a transformer that intentionally learns a smooth decision boundary.

**Strengths:**

Fascinating, insightful paper. Well-written, good visualizations.
Robustly evaluated across model families, model sizes, number of examples. Even examines the effects of quantization on the decision boundary, which might be one of the most important points in the paper (4-bit quantization affects the boundary quite a bit).
Includes sections on fine-tuning to smooth decision boundary as well as training from scratch to do so.
Great RW section, up to date with the most recent work.
No-brainer accept.

**Weaknesses:**

None that I can see.

**Questions:**

N/A.

**Limitations:**

N/A.

---

> ### Author Rebuttal · Authors · 2024-08-07
>
> Dear reviewer WmaA,
>
> Thank you for the positive and encouraging review! We are glad to find that you found our work to be insightful, well-written and robustly evaluated :)

---

### Official Review · Reviewer_PtXB · 2024-07-25

**Soundness:** 2
**Presentation:** 3
**Contribution:** 2
**Rating:** 5
**Confidence:** 3

**Summary:**

This paper conducts a wide range of experiments exploring the smoothness of decision boundary generated by LLMs. Synthetic datasets from scikit-learn are used for the experiments, and experimental results are showing that various factors affect decision boundaries of LLMs.

**Strengths:**

- There are analyses from diverse perspectives, in terms of model size, in-context examples, quantization, prompt format, example order. Extensive experimental results help interpret the targeted phenomenon. Moreover, changes when fine-tuning LLMs with in-context examples are observed.

**Weaknesses:**

**[W1] Justification of task and dataset**:
- There have been works trying to understand the mechanism of in-context learning by conducting experiments on NLP classification tasks. However, it seems unclear why the authors choose scikit-learn-generated dataset. Is this dataset proper to investigate the effectiveness of in-context learning? In addition, can experimental results using this dataset be generalized to NLP classification tasks?
- Moreover, there are concerns on comparison groups. Traditional ML algorithms (e.g., Decition Tree, K-NN, etc.) map each datum into vector spaces, while  LLMs map the entire context into vectors. Can comparing these two mechanism be regarded as a fair process?

**[W2] Analysis**: I have concerns that there are sets of experimental reporting without interpretations of probable reasons. There is a core question: How the smoothness of decision boundary helps understand in-context learning mechanism?

**(Update) After the author's rebuttal, most of these concerns are clarified, thus raising the score 3 to 5.**

**Questions:**

- It would be better to define the concept of smoothness of decision boundry more formally since expecting a perfect smoothness of LLMs' decision boundary is rather unrealistic.

**Limitations:**

Yes

---

> ### Author Rebuttal · Authors · 2024-08-07
>
> Dear Reviewer PtXB,
>
> Thank you for your feedback. We hope to address your questions and comments below.
>
> > Q1: There have been works trying to understand the mechanism of in-context learning by conducting experiments on NLP classification tasks. However, it seems unclear why the authors choose scikit-learn-generated dataset. Is this dataset proper to investigate the effectiveness of in-context learning?
>
> 1. Our primary motivation for using synthetic 2D classification datasets is their amenability to decision boundary visualization, which is central to our paper's focus as decision boundaries are typically studied in 2D space, and our proposed mechanism offers novel insights into the generalization abilities of in-context learning. NLP classification tasks, with their discrete token nature and variable input lengths, pose significant challenges for visualization of decision boundaries along x and y axes.
> 2. Our use of synthetic function datasets is firmly grounded in established practices for studying in-context learning. Several recent high-profile works have exclusively used synthetic data to investigate ICL/learning algos:
>
>     - NeurIPS 2022 paper (https://arxiv.org/pdf/2208.01066) focuses entirely on linear functions for ICL.
>     - ICLR 2023 paper  (https://arxiv.org/pdf/2211.15661) explores learning algorithms for ICL and primarily conducts experiments on synthetic linear functions.
>     - Google's research (https://arxiv.org/abs/2303.03846) employs synthetic linear function classes (see Section 6).
>     - NeurIPS 2019 paper (https://arxiv.org/pdf/1905.11604) utilizes similar scikit-learn datasets to study decision boundaries for SDE algorithms (see Figure 1).
> These papers demonstrate that our approach is quite prevalent in the field. The use of scikit-learn is simply a means to generate widely accepted and reproducible benchmarks with clear characteristics such as linear/non-linear patterns.
>
> 3. Following up your suggestions, we include additional experiments on 6 nlp classification datasets. Note, to handle the high dimensionality of text embeddings, we projected them onto 2D space using t-sne. (Any dimensionality reduction technique will introduce confounders to the analysis, but this price is inevitable to extend our analysis). We experiment with a total of 6 widely-used NLP classification tasks, with both binary and multi-class settings. These include sentiment analysis and Textual Entailment Recognition. As shown in **Figure 1 of the rebuttal PDF**, we demonstrate similar non-smooth decision boundary characteristics in NLP tasks as observed in our synthetic datasets.
>
>
> > Q2: there are concerns on comparison groups. Traditional ML algorithms (e.g., Decition Tree, K-NN) map each datum into vector spaces, while LLMs map the entire context into vectors. Can comparing these two mechanism be regarded as a fair process?
>
> We view in-context learning in LLM as a learning algorithm and use decision boundary visualization as a tool to analyze the generalization ability. In this sense, ICL in LLM is comparable to the traditional classifiers and MLP. Related works have also studied ICL in LLM as learning algorithms. For example, Akyürek et al. proves transformers can implement learning algorithms for linear models based on GD and closed-form ridge regression.
>
> > Q3: "There is a core question: How the smoothness of decision boundary helps understand in-context learning mechanism?" & "... can experimental results using this dataset be generalized to NLP classification tasks?"
>
> - Our work offers novel insights into the ICL mechanism by analyzing decision boundary characteristics, an aspect previously unexplored. Unlike prior research focused on accuracy metrics, we examine the underlying decision-making process of LLMs during ICL. The smoothness of decision boundaries provides valuable information about the model's generalization capabilities, as every grid point (except the in-context examples) lies outside the “training set.”
>
> - To further link decision boundary smoothness with NLP task performance, we fine-tuned a Llama3 8B model on synthetic tasks and tested it on 7 NLP ICL datasets. As shown in **Table 1 of the rebuttal PDF**, the fine-tuned model shows higher performance on several tasks and improved average performance across all tasks. These results suggest that smoother decision boundaries can contribute to generalization in NLP tasks.
>
> > Q4: ... better to define the concept of smoothness of decision boundry more formally since expecting a perfect smoothness of LLMs' decision boundary is rather unrealistic.
>
> - Formally measuring smoothness is challenging due to grid discretization. Qualitatively, the difference in smoothness is evident when comparing traditional models (e.g., kNN, DT) to LLMs. To address this concern quantitatively, we explored several metrics, including curvature (average rate of change of the tangent vector along the decision boundary) and boundary length. However, measuring the continuity behavior of decision boundaries with discretized grids is non-trivial and these metrics are challenging to define and often do not show obvious trends.
> - Nevertheless, we define an empirical metric for decision boundary smoothness: Nearest Neighbor Entropy. This calculates the average entropy of predictions among the k-nearest neighbors for each grid point.
> - We show in **Table 2 of the rebuttal PDF** the NN entropy for the models shown in Figure 1 of our submitted paper, comparing the smoothness of traditional models and sota LLMs. LLMs exhibit higher entropy than models like Decision Trees and kNN. We also provide results for different LLMs with 64 and 128 in-context examples, showing that as the number of examples increases, the entropy generally decreases. This quantitative approach, combined with our qualitative observations, provides a more comprehensive analysis of decision boundary.
>
> We hope this addresses your concerns. Please let us know if you have any questions.

---

> ### Author Response · Authors · 2024-08-12
>
> Dear Reviewer PtXB,
>
> Thank you again for your helpful feedback! Your suggestions have helped us solidify our experiments and analysis by:
>
> 1. Conducting additional experiments on NLP tasks spanning 6 NLP classification datasets,
> 2. Demonstrating that smoothness on synthetic datasets can lead to improvements on downstream tasks, on average, across 7 ICL NLP tasks, and
> 3. Using quantitative metrics: nearest neighbor entropy, as a measure of smoothness.
>
> We hope this addresses the concerns you listed in your review. As the discussion period is nearing its end, could you please let us know if you have any further questions for us? We are happy to address any questions or concerns you may have. Thank you for you time!

---

> > ### Comment · Reviewer_PtXB · 2024-08-13
> >
> > Thank you for clarifying most of my concerns. I understand synthetic benchmarks are also used in this analytic field. Furthermore, I would appreciate for including NLP experiment results. It clearly helps my understanding.
> >
> > I believe refining smoothness can be a key to understand and improve LLMs, but I'm still doubtful on the hypothesis that smoothness has a causal relationship with model performance by relying on observed case samples. Thus, I would like to raise my score from 3 to 5.

---

> > > ### Author Response · Authors · 2024-08-13
> > >
> > > Thank you for raising the score and for your valuable feedback! We appreciate your insights on smoothness and are glad the NLP experiments clarified your understanding. We will incorporate the analysis of these results into our paper. Thank you!

---

### Author Rebuttal · Authors · 2024-08-07

To address the reviewer's concerns, we conducted additional experiments. Here is a summary:

- **NLP Multi-class Classification Tasks (Figure 1):**

We included additional experiments on six widely-used NLP classification datasets, addressing concerns about our use of toy datasets. Note, to handle the high dimensionality of text embeddings, we projected them onto 2D space using t-sne. (Any dimensionality reduction technique will introduce confounders to the analysis, but this price is inevitable to extend our analysis). We experiment with a total of 6 widely-used NLP classification tasks, with both binary and multi-class settings. These include sentiment analysis and Textual Entailment Recognition. As shown in Figure 1 of the rebuttal PDF, we demonstrate similar non-smooth decision boundary characteristics in NLP tasks as observed in our synthetic datasets.

- **Impact of Smoothness on NLP Performance (Table 1):**

Following up reviewer’s concerns in how decision boundary smoothness affect NLP task performance, we fine-tuned a Llama3 8B model on synthetic classification tasks and tested it on 7 diverse NLP datasets on ICL. As shown in the Table 1 of the rebuttal pdf, the results were promising: although only fine-tuned on synthetic classification dataset, the fine-tuned model showed significantly higher performance on several NLP ICL tasks and has an overall improved average performance across 7 tasks, suggesting smoother boundaries in synthetic tasks can contribute to better generalization in NLP tasks.

- **Quantifying Decision Boundary Smoothness (Table 2):**

Following up reviewer’s concerns on the lack of formal definition of smoothness of decision boundary in our setting. We define an empirical metric for decision boundary smoothness: Nearest Neighbor Entropy. This calculates the average entropy of predictions among the k-nearest neighbors for each grid point. Our analysis reveals that LLMs exhibit higher entropy than traditional models like Decision Trees and kNN. Furthermore, increasing the number of in-context examples generally decreases entropy, providing quantitative support for our qualitative observations. Apart from this metric, we also explored several other metrics, including curvature (average rate of change of the tangent vector along the decision boundary) and boundary length (total length of the decision boundary).  However, measuring the continuity behavior of decision boundaries with discretized evaluations is non-trivial and these quantitative metrics are challenging to define and often do not show obvious trends.

- **Influence of Prompt Format (Figure 2):**

We find that the LLM's decision boundary is also affected by the prompt format, which aligns with the importance of prompt engineering in ICL. However, the overall non-smoothness level of the decision boundaries remained consistent. We view prompt formats as another factor influencing the decision boundary rather than undermining our results, since our central observations regarding the non-smoothness of decision boundaries show similar patterns across different prompts. We will add this as another influencing factor in our paper.

---

### Decision · Program_Chairs · 2024-09-25

**Decision:**

Accept (poster)

**Comment:**

This paper proposes a new way to interpret in-context learning for binary classification by analyzing the decision boundaries. The work shows that these boundaries frequently are not smooth, examines the reasons for this and proposes methods for improving the generalizability of in-context learning. Authors perform additional experiments on 6 NLP datasets/tasks based on reviewer suggestions. Rebuttals result in reviewers increasing their scores.